# Including population and environmental dynamic heterogeneities in continuum models of collective behaviour with applications to locust foraging and group structure

**Fillipe Georgiou**[1]*, **Camille Buhl**[2], **J. E. F. Green**[3], **Bishnu Lamichhane**[4], **Ngamta Thamwattana**[4]

**1** Institute for Mathematical Innovation, University of Bath, Bath, United Kingdom, **2** School of Agriculture, Food and Wine, University of Adelaide, Adelaide, South Australia, Australia, **3** School of Computer and Mathematical Sciences, University of Adelaide, Adelaide, South Australia, Australia, **4** School of Information and Physical Sciences, University of Newcastle, Callaghan, New South Wales, Australia

* fg543@bath.ac.uk

**Data availability statement:** All relevant data and code are within the manuscript,

## Abstract

Collective behaviour occurs at all levels of the natural world, from cells aggregating to form tissues, to locusts interacting to form large and destructive plagues. These complex behaviours arise from relatively simple interactions amongst individuals and between individuals and their environment. For simplicity, mathematical models of these phenomena often assume that the population is homogeneous. However, population heterogeneity arising due to the internal state of individuals can affect these interactions and thus plays a role in the dynamics of group formation. In this paper, we present a partial differential equation model that accounts for this heterogeneity by introducing a state space that models an individual's internal state (*e.g.* age, level of hunger) which affects its movement characteristics. We then apply the model to a concrete example of locust foraging to investigate the dynamic interplay of food availability, hunger, and degree of gregarisation (level of sociability) on locust group formation and structure. We find that including hunger lowers group density and raises the percentage of the population that needs to be gregarious for group formation. Within the group structure itself we find that the most gregarious and satiated locusts tend to be located towards the centre with hunger driving locusts towards the edges of the group. These two effects may combine to give a simple mechanism for locust group dispersal, in that hunger lowers the group density, which in turn lowers the gregarisation, further lowering density and creating a feedback loop. We also note that a previously found optimal food patch size for group formation may be driven by hunger. In addition to our locust results, we provide more general results and methods in the attached appendices.

Supporting Information files, Zenodo (DOI: 10.5281/zenodo.11613736) and a linked gitlab repository.

**Funding:** FG received a Lift-Off Fellowship from the Australian Mathematical Sciences Institute for this work (https://austms.org.au/award-and-grant/lift-off-fellowships/). All other authors received no specific funding for this work. The publication fee was covered under The University of Bath PLOS Flat Fee Arrangement (https://library.bath.ac.uk/open-access/publishing-deals) The funders had no role in study design, data collection and analysis, decision to publish, or preparation of the manuscript.

**Competing interests:** The authors have declared that no competing interests exist.

## Author summary

Collective behaviour occurs at all levels of the natural world, from cells joining together to form complex structures, to locusts interacting to form large and destructive plagues. We can model these complex behaviours using mathematics. However, we often need to rely on simplifying assumptions to keep the mathematics easy enough to analyse. One simplifying assumption that is often employed is assuming that all the modelled organisms are the same (or in one of only a few possible states). However, this is often not the case in nature where the differences between individuals arising due to internal characteristics, such as hunger or age, often affect their behaviour and thus can change group dynamics. In this paper we introduce a mathematical model that is able to capture these differences and apply the newly developed model to locust foraging. We find that hunger tends to drive individuals to the edges of aggregations as well as lowers the maximum possible density. These two results combine to give a possible mechanism for group disintegration. Finally, we see a reemergence of an optimal food patch size for group formation, in this instance driven by hunger.

## 1. Introduction

Complex macroscopic self-organised collective motion and pattern formation can emerge from relatively simple interactions amongst individuals, and between individuals and their environment [1–3]. Examples of such behaviour occur at all scales in the natural world, from cells aggregating to form tissues [4], to locusts interacting to form large and destructive plagues [5]. Mathematical models have proven to be an important tool in studying these phenomena [6]. One key assumption frequently made in the interest of mathematical tractability is that the population of individuals is homogeneous; that is, individuals are indistinguishable from each other or only assume a small number of discrete states, e.g. solitarious/gregarious locusts [7,8]. However, in real populations individuals show heterogeneity in attributes such as age, size, personality, and/or fast-changing factors such as energy reserves. For example, hungrier individuals tend to migrate towards the front of moving groups in both caterpillars [9] and crimson spotted rainbow-fish [10], while faster pigeons hold a leadership position in flocks [11]. The aim of this paper is to present a continuum modelling framework in which heterogeneities within a population can be accounted for, and apply it to the important example of locust group formation to understand the role of hunger level in this process.

Locusts are short horned grasshoppers that exhibit density-dependent phase-polyphenism (i.e., two or more distinct phenotype expressions from a single genotype depending on local population density) [12]. In locusts there are two key distinct phenotypes, solitarious and gregarious, with the process of transition from solitarious to gregarious called gregarisation (and the reverse, solitarisation). Gregarisation affects many aspects of locust biology from colouration [13], to reproductive features [14], to behaviour [15]. Behaviourally, solitarious locusts are characterised by an active avoidance of other locusts, whereas gregarious locusts are strongly attracted to each other. Gregarisation is brought about by locusts crowding together and can be reversed by isolating the individuals [16]. In the Desert locust (*Schistocerca gregaria*), gregarisation can take as little as 4 hours with the time-frame for reversal dependent on the length of time the individual has been gregarious (again, potentially as little as 4 hours) [12].

It is in the gregarious state that adult locusts form the infamous large and destructive swarms during plagues. Prior to this, juvenile locusts form hopper bands, large groups sometimes numbering millions of individuals, marching in unison [16]. Existing mathematical models of locust movement aim to capture the onset of collective behaviour by modelling the phase polyphenism as two distinct sub-populations, with density-dependent rates of transition between the two [7,8]. Within the solitarious and gregarious classes, the individuals are all assumed to be identical. However, observations of locust behaviour in experiments suggest that the degree of gregarisation might be depicted more accurately as a spectrum, rather than a binary solitarious/gregarious distinction [17]. Furthermore, their movement characteristics can vary dependent on a variety of factors: for example, locusts modify their dispersal behaviour based on their level of hunger [18–20]. It is how these interactions of internal state and environment affect group formation that we aim to understand using our model.

Mathematical models for locust movement, and for collective behaviour more generally, can be broadly classified as either discrete (where organisms are modelled as individuals) or continuum (in which they are represented using a continuous population density function) [6]. A frequently used discrete approach is the self-propelled particle (SPP) model, in which organisms are modelled as individual points which update their velocity according to prescribed interaction rules. SPPs can be further categorised as second order models, if they include particle inertia, and first order (or kinematic) if inertia is neglected [21]. Second order SPP models have been used fairly extensively to describe locust behaviour (amongst other applications) where they capture the commonly seen collective movement mechanisms such as alignment or pursuit/escape interactions [22–25]. First order SPP models, where drag is assumed to dominate over inertia, are less common in locust modelling but have been used to model disordered group behaviour such as the rolling locust swarm [27,28].

A key drawback of discrete models is that there are few analytical tools available to study their behaviour. In contrast, continuum or partial differential equation (PDE) models, in which organisms are represented as a population density that is a function of space and time, can be analysed using an array of tools from the theory of PDEs. They are most appropriately used when there are a large number of individuals since they do not account for individual behaviour, instead giving a representation of the average behaviour of the group. Given the large numbers of locusts generally present in hopper bands, in this paper we adopt a PDE based approach based on the non-local aggregation equation, first proposed by Mogilner and Edelstein-Keshet to model swarming behaviour [29], and more recently used for modelling locusts [7,8]. This is a PDE analogue of the kinematic SPP model [30,31]. It consists of a conservation of mass equation for the population density $\rho(x, t)$ given by

$$\frac{\partial \rho}{\partial t} + \nabla \cdot \left[ -\nabla \left( Q * \rho \right) \rho \right] = 0, \tag{1}$$

where $Q$ is defined as a social interaction potential and $*$ is the convolution operation. For this type of model, the existence and stability of swarms has been proven both as stationary aggregations and as travelling waves [29]. In addition, analytic expressions for the steady states can been found [28]. This model has been further extended to include non-linear local repulsion which leads to compact and bounded solutions [32] (i.e., solutions where the organisms are contained within a finite area and the population density cannot become arbitrarily large). Whilst this approach has been most frequently used for single populations, the model has also been extended to consider multiple interacting populations [33]. However, although numerous examples of modelling heterogeneity using SPP models exist (see e.g., [6]

and references therein) there are few examples of continuum models [34], and none (to our knowledge) that include both phenotypic and environmental heterogeneity.

In this paper, we first present a general model of collective behaviour that includes dynamic and continuous population heterogeneity and environmental interactions in Sect 2 (and S1 Appendix) and adapt it to locust foraging behaviours. We then obtain analytic results of maximum density and the conditions required for the formation of aggregations under the assumption of homogeneity in Sect 3.1 (with more general results in S2 Appendix). We then look, in Sect 3.2, at numerical simulations of locust foraging and investigate the effect of a continuous model of gregarisation and hunger on group formation and structure within spatially isolated food patches. Finally, we discuss our results and further avenues of exploration in Sect 4.

## 2. Mathematical model

We begin by presenting a continuous kinematic model of collective behaviour that includes both local and non-local inter-individual interactions, as well as environmental interactions, where these interactions may be mediated by the internal state of the organism. For simplicity, we consider a single population, which is represented as a density of individuals (number per unit area), $\rho$, at point $\boldsymbol{x}$ in space, at time $t$, and with internal state $\boldsymbol{n}$. Here, $\boldsymbol{n} = (n_1, .., n_N)$ is an $N$-dimensional continuous variable, where each state $n_i$ ($i = 1, \ldots, N$) can take a value on the interval $[0, 1]$. The states can be used to represent a range of characteristics of individual organisms, such as their degree of satiation, age, stage of gregarisation, *etc.*, as a fraction of the greatest possible value, *i.e.*, all the states are normalised and dimensionless. We will term the combination of all possible internal states the state space, $\Omega_{\boldsymbol{n}}$ (*i.e.*, the state space is an $N$-dimensional unit hypercube).

We make the following modelling assumptions:

1. Organisms can be classified by their internal state and this state is continuous in nature.
2. Environmental interactions are local in nature (*i.e.,* an organism at point $\boldsymbol{x}$ is only influenced by the environment at point $\boldsymbol{x}$). (Whilst it is plausible that there could be long-range environmental influences, *e.g.* mediated by sight, sound or smell, we restrict our attention to short-ranged interactions here for simplicity.)
3. Local interactions that directly affect movement between organisms are repulsive (*i.e.,* individuals try to avoid collisions or close physical contact).
4. Organisms also experience a non-local (*i.e.,* longer-ranged) interaction (this could include both attractive or repulsive interactions mediated by longer-range sensing such as sight, hearing or smell).
5. The nature of all interactions depends upon the organism's internal state and local environmental conditions, and not by those of other organisms.

Since the density of organisms in space, state, and time is given by $\rho(\boldsymbol{x}, \boldsymbol{n}, t)$, the total local density of organisms at a point $\boldsymbol{x}$ and time $t$ is defined as:

$$\bar{\rho}(\boldsymbol{x}, t) = \int_{\Omega_{\boldsymbol{n}}} \rho(\boldsymbol{x}, \boldsymbol{n}, t) d\boldsymbol{n}, \tag{2}$$

For later convenience we will also define $\Omega_{\boldsymbol{x}}$ as our spatial domain and the total mass of organisms at time $t$ as:

$$M(t) = \int_{\Omega_{\boldsymbol{x}}} \bar{\rho}(\boldsymbol{x}, t) \, d\boldsymbol{x}. \tag{3}$$

We do note that in the case of this paper, the mass is conserved, i.e. $M(t) = M$ a constant. Finally, we introduce a generic variable, $E$, to describe the environment. This might represent such environmental conditions as food density, sunlight, temperature, etc. In general, $E$ may be a vector of values that change in space and time, and as a result of interactions with organisms. However, for simplicity, henceforth we consider only a single environmental variable.

We assume that the time-scales we are investigating are shorter than the life cycle of the organism, ignoring births and deaths and thus conserving the total number of organisms. We consider local organism-organism interactions (e.g. crowding), local organism-environment interactions (e.g. foraging), and non-local organism-organism interactions (e.g. sight and smell mediated interactions between individuals). We also allow for a continuous change within the state space (e.g. the change in an organism's hunger level). Hence, conservation of organisms gives an equation of the form:

$$\frac{\partial \rho}{\partial t} + \nabla \cdot \left( J_{\text{local}} + J_{\text{non-local}} \right) + \nabla_{\boldsymbol{n}} \cdot \left( J_{\text{state}} \right) = 0, \tag{4}$$

where $J_{\text{local}}$ is the flux due to local interactions, $J_{\text{non-local}}$ is the flux due to non-local interactions, $J_{\text{state}}$ is the flux within the state space and $\nabla_{\boldsymbol{n}}$ represents the differential operator, $\nabla$, applied to the state space variables, i.e.

$$\nabla_{\boldsymbol{n}} = \left( \frac{\partial}{\partial n_1}, ..., \frac{\partial}{\partial n_N} \right). \tag{5}$$

For the local interactions, we follow the work of Painter and Sherratt [35] (who derive them as the limit of a lattice model - see S1 Appendix for details ) and assume that organism movement due to local interactions depends on local environmental conditions, the state of the organism, and local population density. These assumptions give our local flux as:

$$J_{\text{local}} = -D \left[ \nabla \left( f_l(\boldsymbol{n}, E) \rho \right) + \gamma \rho \left( f_l(\boldsymbol{n}, E) \nabla \tau(\bar{\rho}) \right) \right], \tag{6}$$

where $D$ is the linear diffusion coefficient and $\gamma$ is parameter which specifies the strength of the advective flux. We have introduced the function $f_l(\boldsymbol{n}, E)$, which relates the state and local environmental conditions to the local aspect of movement: it can be thought of as specifying how the mobility of the organisms depends on local conditions. Similarly, the function $\tau(\bar{\rho})$ specifies how the organisms' movement is affected by population density gradients. (We will give concrete examples of these functions when applying the model to locust behaviour in the following subsection.)

Next, for our non-local interactions, we adopt the fluxes used originally by Mogilner and Edelstein-Keshet [29] and assume that the movement of each organism depends on the distance-weighted sum of the surrounding population density. We introduce $Q(\boldsymbol{x})$ , which is termed the social potential and gives the distance weighting for non-local interactions. We then introduce the function $f_n(\boldsymbol{n}, E)$ which specifies how the strength and direction of the social potential depends on the internal state and environmental conditions. These assumptions give:

$$J_{\text{non-local}} = -f_n(\boldsymbol{n}, E) \nabla \left( Q(\boldsymbol{x}) * \bar{\rho}(\boldsymbol{x}, t) \right) \rho(\boldsymbol{x}, \boldsymbol{n}, t), \tag{7}$$

where $*$ is the convolution operation. A heuristic derivation of the non-local flux can be found in S1 Appendix.

Finally, we assume that organisms change their internal state based on some $N$-dimensional state vector field, $\boldsymbol{v}_n$, and that this state evolves continuously in time. This gives our flux as,

$$\boldsymbol{J}_{\text{state}} = \boldsymbol{v}_n \rho, \tag{8}$$

with each element of $\boldsymbol{v}_n = (\dot{n}_1, ..., \dot{n}_N)$ being a function describing the rate of change in state. We note that $\boldsymbol{v}_n$ itself can depend on local environmental conditions, organism density, etc.

Hence, combining 6, 7, 8 and then rewriting as an advection-diffusion equation so that the advective component provided by $f_l(\boldsymbol{n}, E)$ becomes more explicitly obvious. we get the following equations:

$$\frac{\partial \rho}{\partial t} + \nabla \cdot (\boldsymbol{v}_x \rho) + \nabla_{\boldsymbol{n}} \cdot (\boldsymbol{v}_n \rho) = D \nabla \cdot \left[ f_l(\boldsymbol{n}, E) \nabla \rho \right], \tag{9}$$

with

$$\boldsymbol{v}_x = -f_n(\boldsymbol{n}, E) \nabla (Q * \bar{\rho}) - D \left[ \nabla f_l(\boldsymbol{n}, E) + \gamma f_l(\boldsymbol{n}, E) \nabla (\tau(\bar{\rho})) \right]. \tag{10}$$

In order to close our model, we need to specify the functions: $f_l(\boldsymbol{n}, E)$, $Q(\boldsymbol{x})$, $f_n(\boldsymbol{n}, E)$ and $\boldsymbol{v}_n = (\dot{n}_1, ..., \dot{n}_N)$. The appropriate forms for these will depend upon the application being considered, so we consider a concrete example in the following section.

## 2.1. Specialisation to locusts

We now take our general model and tailor it to investigate the effect of food, hunger, and gregarisation on locust group formation. We begin by giving expressions for $\tau(\bar{\rho})$ and $Q$, before stating our environmental function and specifying state dimensions and parameters for gregarisation and hunger.

To begin, we assume that $\tau(\bar{\rho})$, the local aspect of advective movement, is repulsive and proportional to the likelihood of collisions between locusts. We further assume that the probability of collision is independent of the state of the locusts and depends only on the local density. Using the law of mass action we get,

$$\tau(\bar{\rho}) = \bar{\rho}^2. \tag{11}$$

Next, for the social potential, $Q(\boldsymbol{x})$, we adopt the Morse potential used by [36]

$$Q(\boldsymbol{x}) = e^{-\frac{|\boldsymbol{x}|}{r}}, \tag{12}$$

with $r$ representing the average sensing distance of locusts. We now introduce the environment as well as the state dimensions of hunger-satiation and gregarisation.

**2.1.1. The environment.** In addition to the locust density, we include food resources in our model. Let $E = c(\boldsymbol{x}, t)$ denote the food density (mass of edible material per unit area). We assume that locust food consumption is based on their contact with food and their state, and on the time-scale of group formation food production is negligible, giving

$$\frac{\partial c}{\partial t} = -c(\boldsymbol{x}, t) \int_{\Omega_n} \psi(\boldsymbol{n}) \rho(\boldsymbol{x}, \boldsymbol{n}, t) d\boldsymbol{n}, \tag{13}$$

where $\psi(\boldsymbol{n})$ is a function relating the locust's state and food consumption rate.

Food is assumed to affect locust movement only through the local interaction term, $f_l$. For simplicity, we assume this term is a product of two functions, which specify how movement depends on food availability ($f_c(c)$), and how it depends on hunger ($f_{lh}(n_h)$), where $n_h$ is a measure of the degree of hunger of the locusts. (The state variable $n_h$ is explained in more detail below (see State 2), where we also specify the the functional form of $f_{lh}(n_h)$).) Thus,

$$f_l(\boldsymbol{n}, E) = f_l(\boldsymbol{n}, c) = f_c(c)f_{lh}(n_h).\tag{14}$$

For the effect of food on locust movement, we assume that movement decreases as food availability increases and the sensitivity of this effect decreases with increasing food availability [37], with an infinite amount of food leading to no movement. We thus require a monotonically decreasing function of $c$ that approaches 0 as $c \to \infty$, so here we use (inline with [8])

$$f_c(c) = e^{-\frac{c}{c_0}},\tag{15}$$

where $c_0$ relates food amount to pause duration. We then let $c = c_0 c'$ so that pause duration will depend only on the amount of food present, substituting into (15) and dropping the $\cdot'$ notation gives us

$$f_c(c) = e^{-c}.\tag{16}$$

Behaviourally, this corresponds to locusts pausing to eat when food is available, with each doubling of food density having a decreasing effect on the pause duration [38].

**2.1.2. State 1: Gregarisation.** We will denote the dimension of gregarisation as $n_g$. We assume that locusts become increasingly gregarious as the local population density increases (and solitarious with decreasing density). From Anstey et. al. [17], locust gregarisation is proportional to the amount of serotonin in their system, and this serotonin is released either through mechanosensory pathways (collisions) or cephalic pathways (sight and smell). We assume that this release is proportional to their local density but is bounded, and the serotonin decays at some constant rate. Thus we set:

$$\dot{n}_g = \left(f(\bar{\rho}) - k n_g\right),\tag{17}$$

where, $f(\bar{\rho})$ is a positive monotonically increasing bounded function representing normalised density dependant serotonin release rate, and $k$ represents the normalised decay rate of serotonin in the locusts system. For our function $f(\bar{\rho})$ we select:

$$f(\bar{\rho}) = \frac{\delta\left(\frac{\bar{\rho}}{\kappa}\right)^2}{1 + \left(\frac{\bar{\rho}}{\kappa}\right)^2},\tag{18}$$

where $\delta$ is maximal normalised serotonin release rate and $\kappa$ is the locust densities at which half this maximal rate occurs (this is the same function as that for the rate of gregarisation used by Topaz et. al. [7]). We note that we are only modelling the initial gregarisation process and this equation is not valid for locusts that have been gregarious for an extended period of time, i.e., it represents acute and not chronic gregarisation [39].

We then need to specify how gregariousness affects the movement of the locusts. We assume that this only affects the non-local component of movement (*i.e.*, the function $f_n$), and that the interaction is repulsive when the locusts are solitarious, and attractive when they are gregarious. We let $n_g = 0.5$ be the transition point between solitarious and gregarious

behaviour, and assume that the transition from repulsion to attraction is fairly 'sharp'. We thus model the effect of gregariousness on movement through the sigmoid function

$$f_n(\boldsymbol{n}, E) = f_n(n_g) = A\left(1 - \frac{2}{1 + e^{-15(n_g - 0.5)}}\right), \tag{19}$$

where $A$ is the maximum strength of the interaction.

**2.1.3. State 2: Hunger-satiation.** We now define the state dimension of hunger-satiation, denoted $n_h$, with $n_h = 0$ being completely starving and $n_h = 1$ being completely satiated. We assume that locusts become hungrier based on energy loss due to metabolism, and that locusts become satiated by eating at a rate proportional to their state. These assumptions lead to the equation

$$\dot{n}_h = \lambda(n_h)c(\boldsymbol{x}, t) - \nu n_h, \tag{20}$$

where $\lambda(n_h)$ describes how fast the locusts eat based on hunger, and $\nu$ is the energy lost due to metabolism. We assume that hunger only affects the local movement term, with hungry locusts moving three times as fast as satiated ones [18] and that the relationship is linear. Finally, we assume that parameter estimations correspond to a locust in hunger state $n_h = 0.5$. This gives the form of $f_{lh}$ from (14) as,

$$f_{lh} = 1.5 - n_h. \tag{21}$$

For our functions of consumption, $\lambda(n_h)$ and $\psi(\boldsymbol{n})$, we assume that the rate changes linearly and that a starving locust eats twice as fast as a satiated one, which gives:

$$\lambda(n_h) = \eta_l(2 - n_h) \tag{22}$$

and

$$\psi(n_h) = \eta_f(2 - n_h) \tag{23}$$

Finally, for all state dimensions we use no-flux boundary conditions, i.e.,

$$\dot{n}_{g,h}\big|_{0,1} = 0.$$

This completes the specification of the model in this case. For convenience, a full list of functions presented in this section can be found in Table 1.

**Table 1. Locust behavioural functions.**

| Function | Description |
| --- | --- |
| $f_{ng}(n_g) = A\left(1 - \frac{2}{1 + e^{-15(n_g - 0.5)}}\right)$ | Function relating gregariousness to non-local interactions |
| $f_{lh}(n_h) = 1.5 - n_h$ | Function relating hunger to non-local interactions |
| $f_c(c) = e^{-c}$ | Function relating food density to local interactions |
| $f_l(\boldsymbol{n}, E) = f_{lh}(n_h)f_c(c)$ | The effect of environment and state on mobility |
| $\lambda(n) = \eta_l(2 - n)$ | The effect of locust feeding on satiation |
| $\psi(n) = \eta_f(2 - n)$ | The amount of food consumed based on locust hunger |

## 3. Results

### 3.1. Analytical results

We now gain some insight into the models behaviour through PDE analysis techniques. We begin by using gradient flow methods [40] to find analytic expressions for the maximum density of aggregations with environmental conditions that are constant in both space and time, as well as homogeneous in organism state. We then use linear stability analysis to look at the conditions required for aggregations to form, first with all organisms being in the same state, and then in one of two states. Whilst these assumptions remove the effects of the heterogeneity of our locusts and their environment, the results provide a baseline from which we can gain insights into the behaviour under more heterogeneous conditions. Whilst we concentrate on the application to locusts here, the complete calculations for a more general model and associated conditions can be found in S2 Appendix.

**3.1.1. Density of aggregations.**   In order to facilitate analysis, we must first introduce some simplifying assumptions. We can then estimate the maximum density and width of aggregations in both the large and small mass limits in one dimension (i.e., as $M \to \infty$ and $M \to 0$, respectively). Specifically, our assumptions are that the food, $c$, is constant in space and time, and all the locusts are in the same state. We consider this state to be changing considerably slower than the movement of the locusts, i.e., it is a pseudo-steady state. Finally, while the linear diffusion causes locusts to be present everywhere in the domain, we will assume that the bulk of locusts are contained within a single aggregation, and label the width or 'footprint' of this aggregation as the support, $\Omega$.

We begin by investigating the maximum density, $||\bar{\rho}||_{\infty}$, and length of the support, $||\Omega||$, as $M \to \infty$. This is done by further assuming that $\bar{\rho}(x)$ is approximately rectangular. In addition, for a single aggregation we assume that the support is far larger than the range of $Q$. We thus approximate $Q \approx V_Q \delta(x)$, where $\delta(x)$ is the Dirac delta function, with $V_Q = \int Q d\boldsymbol{x}$. We can then find our maximum density as

$$||\bar{\rho}||_{\infty} = -\frac{3V_Q}{8D\gamma}F + \sqrt{\left(\frac{3V_Q}{8D\gamma}\right)^2 F^2 - \frac{3}{2\gamma}}, \tag{24}$$

with support

$$||\Omega|| = \frac{M}{||\bar{\rho}||_{\infty}} = \frac{M}{-\frac{3V_Q}{8D\gamma}F + \sqrt{\left(\frac{3V_Q}{8D\gamma}\right)^2 F^2 - \frac{3}{2\gamma}}}. \tag{25}$$

where

$$F = \frac{f_n(\boldsymbol{n}, c)}{f_l(\boldsymbol{n}, c)}, \tag{26}$$

is the ratio of our non-local and local effects. We can also use (24) to estimate the parameter $\gamma$, given a maximum density of locusts, $\rho_{\infty}$. We find

$$\gamma = -\frac{6V_Q}{8D\rho_{\infty}}F - \frac{3}{2\rho_{\infty}^2}. \tag{27}$$

We now turn to the case where $M \to 0$. We assume that for a single aggregation we can approximate the social potential using a Taylor expansion, i.e., $Q(x) = e^{-\frac{|x|}{r}} \approx 1 - \frac{|x|}{r}$. Additionally, we ignore the effect of linear diffusion within $\Omega$ (to make the calculations possible),

giving our maximum density, $||\bar{\rho}||_\infty$, and aggregation support, $||\Omega||$, as

$$||\bar{\rho}||_\infty = \sqrt[3]{\frac{-3M^2F}{8Dr\gamma}}, \tag{28}$$

and

$$||\Omega|| = B\left(\frac{2}{3}, \frac{1}{2}\right) \sqrt[3]{\frac{MDr\gamma}{-3F}}, \tag{29}$$

where $B$ is the $\beta$-function (for definition see [41], page 207). This gives a similar relationship to the large mass limit. However, here both the maximum density and support scale with mass, $M$.

From these results we can see that hunger acts to decrease the maximum density of locusts by increasing their dispersal while the presence of food has the opposite effect (this relationship is inverted for the size of the support). In addition, increasing gregarisation and attraction strength increases the maximum density of locust groups. However, the effect is more pronounced for large groups than for small groups.

We can check the accuracy of our estimates by comparing them to simulation results of our full system of equations (Details of the numerical scheme can be found in S3 Appendix [42–45]). We begin by letting $r = 1$, $A = 1$, $D = 0.01$, and $\gamma = 60$. Then, for each simulation we place a mass of locusts, $M$, in every combination of the states $n_g = 0.85$ (results for $n_g = 0.75$, 0.85 and 0.95 can be seen in S2 Appendix) and $n_h = 0.15$, 0.55, 0.95 in the centre of the domain (we vary the domain size according to mass so there is no boundary interaction) and run it to a pseudo steady state, $t = 1000$.

The results for $||\bar{\rho}||_\infty$ and $||\Omega||$ can be seen in Fig 1. In the plots, the dotted lines represent the minimum estimate between the small and large mass limits for the density (and maximum for the support). We use the minimum estimate as the small mass limit prediction is lower in the region where $M$ is small, but as the small mass limit increases with $M$ we would assume the results approach the constant large mass limit after the estimates cross over. The solid lines represent simulated results. As $||\Omega||$ is theoretically infinite due to the linear diffusion, for the simulated $||\Omega||$ we select the region for which 98% of the mass, $M$, is contained. This number is somewhat arbitrary but consistent across the estimations. We can see that as $M$ increases the simulated limits approach those given by the theoretical estimates. However, there is some error due to the required simplifications in our estimates, numerical error in the simulations, and our method of approximating $||\Omega||$ from the simulations. We note that the small mass estimates are considerably less accurate than the large mass limits due to ignoring linear diffusion.

In addition, we can look at the individual simulation results. In Fig 2, we see the small ($M = 0.48$) and large ($M = 96.17$) mass limit estimates (left and right plots respectively) with their corresponding simulations for $n_g = 0.95$, $n_h = 0.15$, 0.55, 0.95. In the plots, the vertical dotted lines are the estimates of the support, the horizontal dotted lines are the estimates of the maximum density, and the solid lines are the simulation results. We can see that the large mass estimates are considerably more accurate but the small mass estimates do capture some of the qualitative behaviours. For example, that hungry locusts have a lower density than their satiated counterparts.

### 3.1.2. Linear stability analysis of homogeneous steady states.

In order to gain insights into the conditions under which aggregations can form, we investigate the stability of spatially-homogeneous steady states. In this analysis we perturb the homogeneous steady state, $\rho_c$, by adding a small amount of noise. We then find under what conditions (given in

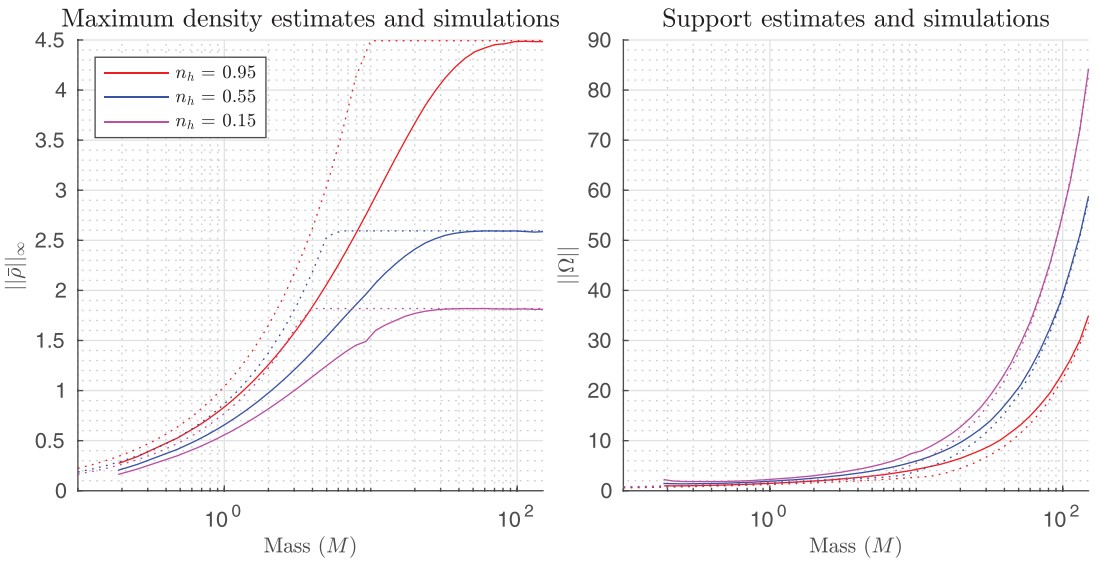

**Fig 1. Small and large mass max density and support, estimates and simulations.** Estimates and simulations for the maximum density (left) and size of the support (right). Estimates for different states are plotted using dotted lines, with the corresponding simulation results plotted in solid lines. As $\|\Omega\|$ is theoretically infinite due to the linear diffusion, for the simulated $\|\Omega\|$ we select the region for which 98% of the mass, $M$, is contained.

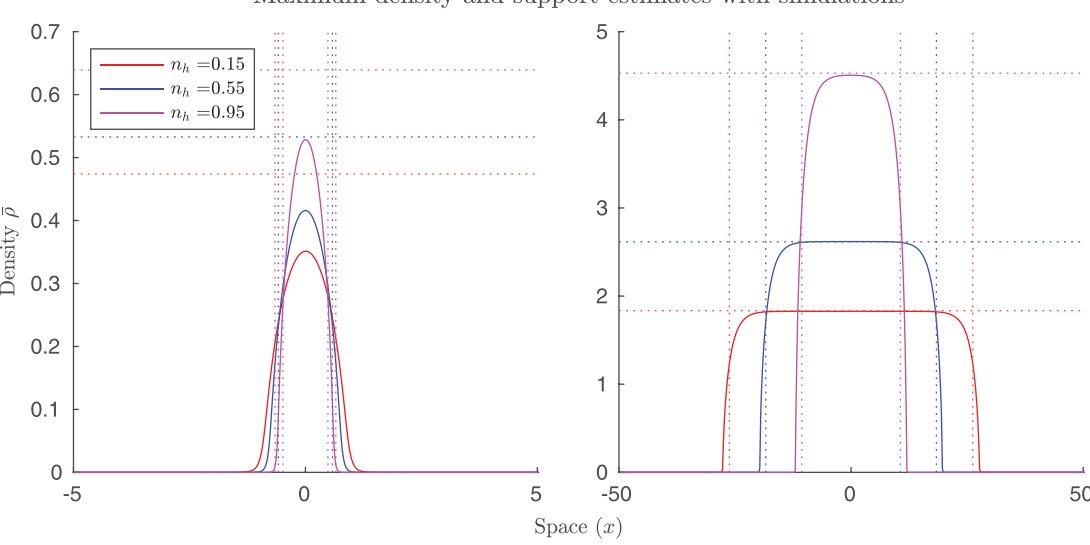

**Fig 2. Individual small and large mass simulations with estimations.** Simulations of small ($M = 0.48$) and large ($M = 96.17$) mass limits (left and right plots respectively). Here, $n_g = 0.95$, and $n_h = 0.15$, $0.55$, and $0.95$. In the plots, the vertical dotted lines are the estimates of the support, the horizontal dotted lines are the estimates of the maximum density, and the solid lines are the simulation results.

the form of an inequality) the small perturbations grow and are likely to lead to aggregations. Note that here, we again assume that $c$ is constant in space and time. Full details of the calculations can be found in S2 Appendix.

We begin by considering the situation where all locusts are in a single state, $\boldsymbol{n}_1$, and assume that the change in state is considerably slower than the movement of the locusts. These assumptions give a condition for instability in terms of our local and non-local state based forces, as

$$-F > \frac{\left(D\gamma\rho_c + \frac{D}{\rho_c}\right)}{2r}, \tag{30}$$

where $F$ is our ratio of non-local to local effects, given by (26). From this we can see that if $F > 0$, corresponding to a repulsive non-local term or solitarious behaviour, then aggregations will not form. If $F < 0$ then non-local forces associated with gregarisation simply need to be greater than the repulsive local forces for aggregations to form. We also see that as locust satiation increases then the level of gregarisation for group formation to occur decreases. In addition, food decreases the required gregariousness for group formation.

We can also consider two distinct sub-populations, a more gregarious state $\boldsymbol{n}_G$ and more solitarious state $\boldsymbol{n}_S$ with

$$\frac{f_l(\boldsymbol{n}_S,c)}{f_n(\boldsymbol{n}_S,c)} > \frac{f_l(\boldsymbol{n}_G,c)}{f_n(\boldsymbol{n}_G,c)}, \tag{31}$$

(i.e. the ratio of local to non-local forces for $\boldsymbol{n}_G$ is less repulsive than for $\boldsymbol{n}_S$, or even attractive). While we have called our populations solitarious and gregarious the locusts do not need to be specifically in those states. For example, we could consider two solitarious populations with one being slightly less solitarious. The fraction of the population in the more gregarious state $\boldsymbol{n}_G$ is given by $\phi$. This allows us to find the condition for instability in terms of $\phi$ as

$$\phi > \phi^* = \frac{2rf_n(\boldsymbol{n}_S,c) + Df_l(\boldsymbol{n}_S,c)\rho_c + \frac{Df_l(\boldsymbol{n}_S,c)}{\rho_c}}{2rA_n + (D\gamma\rho_c + \frac{D}{\rho_c})A_l}, \tag{32}$$

where

$$A_n = f_n(\boldsymbol{n}_S) - f_n(\boldsymbol{n}_G), \text{ and } A_l = f_l(\boldsymbol{n}_S,c) - f_l(\boldsymbol{n}_G,c),$$

and $\phi^*$ represents the critical gregarious fraction where aggregations begin to form (i.e. if the current gregarious fraction is greater than the critical fraction, then an aggregation is likely to form). Then, using (32) we can see that as the amount of available food increases the fraction $\phi$ required for group formation decreases. Additionally, the numerator in (32) is only in terms of $\boldsymbol{n}_S$, thus as the more solitarious locusts become hungrier (or more solitarious) this will increase the fraction $\phi$ required for group formation, with the presence of food acting to decrease it. Next, the denominator depends on the difference between states, thus if the gregarious state has less local repulsion than the solitarious state this decreases the gregarious mass fraction required for group formation. That is, if all the gregarious locusts are satiated while the solitarious are not this would again decrease the mass fraction required. Finally, as $\bar{\rho}$ increases (or goes to 0) the gregarious fraction required for group formation increases suggesting an upper (and lower) bounds on locust density in order to form groups.

We note that if only food is considered, this reduces to the stability condition found in our previous work [8], and if only the non-local forces are considered it reduces to that of Topaz et al. [7]. In contrast to those previous works, the aggregation process can be brought about not only by increasing the gregarious mass fraction but by changing the internal state or environment of the locusts. For example, given a population that is below the critical mass fraction, $\phi^*$, as it becomes satiated (leading to a decrease in $f_l$) this could trigger the locusts to form

aggregations. Conversely, decreasing satiation or gregarisation could prevent aggregations forming.

Finally, we can find the maximum homogeneous density, $\rho_c$, that aggregations can still form. So taking (32) and substituting $\phi^* = 1$ gives a maximum homogeneous density for aggregations as,

$$\rho_c = -\frac{r}{D\gamma}F + \sqrt{\left(\frac{r}{D\gamma}\right)^2 F^2 - \frac{1}{\gamma}}, \tag{33}$$

where $F$ is given by (26) for the state $\boldsymbol{n}_G$. Rather neatly, if $r = \frac{1}{2}$, corresponding to $\int_{\Omega_x} Q(x)dx = 1$, this becomes

$$\rho_c \approx \frac{2}{3}||\rho||_\infty, \tag{34}$$

where $||\rho||_\infty$ is given by (24), this is similar to the relationship previously derived [8]. However, here this depends on the internal state, so it is possible to avoid aggregations forming by changing the internal state either by decreasing satiation or gregariousness.

## 3.2. Simulations

Through our analytic results we can now derive model parameter estimates, full details of which can be found in S4 Appendix. We begin by noting that for the Desert locust (*S. gregaria*), that full gregarisation takes about 4 hours, and that the inverse takes approximately the same amount of time [12,17]. This gives $k = 0.9986\mathrm{hr}^{-1}$, $\kappa = 4.8293\mathrm{locusts\,m}^{-2}$, and $\delta = 0.9960\mathrm{hr}^{-1}$ . Next, using the assumption of very little purely random movement and that the maximum density of locusts is $\approx 1000\mathrm{locusts\,m}^2$ [25], we find $D = 0.01\mathrm{m}^2\,\mathrm{hr}^{-1}$ and $\gamma = 4.5291\mathrm{locusts}^{-2}$. For our non-local term, we use the sensing radius of locusts being 14cm [25] and the approximate speed of locusts being $216\,\mathrm{m\,hr}^{-1}$ [7] to find $r = 0.14\mathrm{m}$ and $A = 6.83\mathrm{m}^3\,(\mathrm{hr\,locust})^{-1}$. For our metabolism parameters, we note that locusts take one day to reach their maximum starvation (assuming approximately 12 hours of activity) and that 88 min of consumption of a 5 hour period will leave a locust fully satiated [18]. We then find $\nu = 0.2496\mathrm{hr}^{-1}$ and $\eta_l = 0.9362\mathrm{hr}^{-1}$. Finally, to keep our food interactions similar to our previous work we re-dimensionalise the parameter $\kappa$ from our previous study [8], and assume that this corresponds to a hunger level $n_h = 0.5$. A full list of parameters derived in this section can be found in Table 2.

Using the functions and parameters defined in Tables 1 and 2 respectively, we perform numerical experiments to investigate the effect of gregarisation and hunger on group heterogeneity, and the interaction between these states and the distribution of food on group formation. For all simulations, our spatial domain is the interval $x \in [0, 3]$ (this is the same domain used by [7]), with periodic boundary conditions (i.e., $\rho(0, \boldsymbol{n}, t) = \rho(3, \boldsymbol{n}, t)$), for a total of 10 model hours. These experiments are equivalent to those in our previous work [8], and the numerical scheme used is given in S3 Appendix.

The initial locusts density is given by

$$\rho(x, n, 0) = \frac{2\rho_{\mathrm{amb}}}{16.6}(16.6 + \mu)(1 - \mathcal{H}(n_g - 0.5)), \tag{35}$$

where $\rho_{\mathrm{amb}}$ is a ambient locust density, $\mu$ is some normally distributed noise, $\mu \sim \mathcal{N}(0, 1)$, and $\mathcal{H}$ is the Heaviside function, i.e., we have locusts uniformly distributed over the solitarious region of the gregarisation state. In order to ensure that simulations are comparable we

**Table 2. Parameters used in numerical simulations.**

| Variable | Description | Value | Units | Source |
|---|---|---|---|---|
| $k$ | Normalised decay rate of serotonin | 0.9986 | hr$^{-1}$ | [12,17] |
| $\kappa$ | Density of half normalised maximal serotonin release rate | 4.8293 | locusts m$^{-2}$ | [22] |
| $\delta$ | Normalised maximal serotonin release rate | 0.9960 | hr$^{-1}$ | [12,17] |
| $D$ | Linear diffusion coefficient | 0.01 | m$^2$ hr$^{-1}$ | |
| $\gamma$ | Non-linear diffusion coefficient | 4.5291 | locusts$^{-2}$ | (27) [25] |
| $A$ | Strength of non-local interactions | 6.83 | m$^3$ (hr locust)$^{-1}$ | [7] |
| $r$ | Range of non-local interactions | 0.14 | m | [7,25] |
| $\nu$ | Normalised energy lost due to metabolism | 0.2496 | hr$^{-1}$ | [18] |
| $\eta_l$ | Normalised energy gain due to consumption | 0.9362 | hr$^{-1}$ | [18] |
| $\eta_f$ | Normalised food consumption rate | 0.0384 | hr$^{-1}$ | [18] |

set up initial locust conditions and rescale them for each given ambient locust density, i.e., each simulation has the same noise at each point in space.

Finally, the initial condition for food is given by a smoothed step function of the form:

$$c(x,0) = \frac{1}{2\zeta}\left[\tanh\left(\alpha\left[x - \left(x_0 - \frac{\zeta}{2}\right)\right]\right) - \tanh\left(\alpha\left[x - \left(x_0 + \frac{\zeta}{2}\right)\right]\right)\right], \qquad (36)$$

with $\alpha = 21$, $x_0 = 3/2$, and $\zeta$ being the initial food footprint size. We also introduce $\omega = 100\zeta/3$ which expresses the size of the food footprint as a percentage of the domain. This roughly corresponds to isolated food patches surrounded by empty space.

We begin by considering ambient locust densities around the expected onset of group behaviour ($\approx \rho_{\text{amb}} = 5$ [22]) both with and without a hunger dimension (for the case without, $n_h = 0.5$ for all locusts and is unchanging). We vary the ambient density from $\rho_{\text{amb}} = 4.8$ to $\rho_{\text{amb}} = 5.2$ in the case where hunger is absent and from $\rho_{\text{amb}} = 4.8$ to $\rho_{\text{amb}} = 5$ where hunger is present. Finally, we vary the food footprint ranges from $\omega = 5\%$ to $\omega = 50\%$. (We also considered the case where there is no food. However all simulations resulted in no aggregations forming, likely due to there being no symmetry breaking mechanism present within this parameter range, so these results are not shown here and can be found in S3 Appendix.)

The results can be seen in Fig 3, with the absence of a hunger dimension on the left and the presence of a hunger dimension on the right. We find that when only considering gregarisation, narrower food footprints decrease the required density for group formation. In contrast, when we also include hunger we find a greater effect of food and the presence of an optimal food width. We interpret this as arising from the competition between two effects. The presence of food reduces the movement rate of locusts, so when hunger is not considered, small concentrated patches of food will tend to cause locusts to linger on the patch and reach a population density at which gregarisation can occur. However, when hunger is

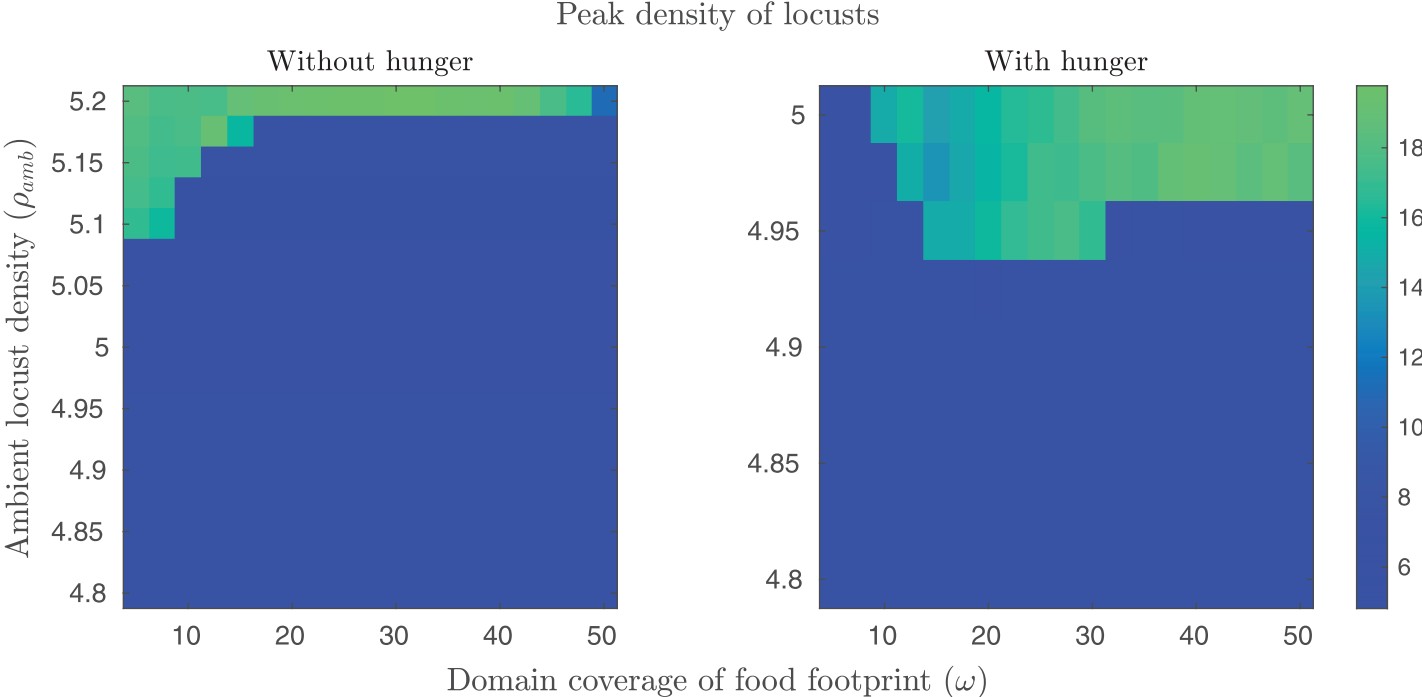

**Fig 3. Maximum locust density with varying food footprint sizes and initial ambient locust densities.** For the simulations, $x \in [0,3]$ with periodic boundary conditions and $t \in [0,10]$. Ambient locust density ranges from $\rho_{amb} = 4.8$ to $\rho_{amb} = 5.2$ in the case where hunger is absent and from $\rho_{amb} = 4.8$ to $\rho_{amb} = 5$ where hunger is present. Food footprint ranges from $\omega = 5\%$ to $\omega = 50\%$. The plots show the maximum locust density at $t = 10$ for the varying food footprint sizes and ambient locust densities with a food mass of 1.

considered, we observe that the hungrier locusts are more strongly repelled by others, causing greater dispersal from small patches as the locust density increases. Similarly, we note that the narrower footprints lead to a reduction in the final peak density when hunger is considered.

By looking at the individual simulations in Fig 4 we can see the effects of both gregarisation and hunger on group heterogeneity. In Fig 4 food is plotted in green, whilst total locust density represented by a line that varies in colour based on the average locust gregarisation at that point, changing from blue when fully solitarious to red when fully gregarious. The line width shows the average hunger, with thin being hungry and thick being satiated. (However, this plotting technique does give rise to an accordion style visual artefact.) This figure illustrates the effect of the optimal food width. When the food is too narrow there is attempted group formation, but there are insufficient numbers of gregarious locusts (or the locusts are insufficiently satiated) so the group does not persist. If the food is too wide groups do not form. Once a group has formed, on average the most gregarious and satiated locusts end up in the middle of the group with a reduction in gregarisation and an increase in hunger towards the edges. In addition, the most gregarious appear to be the most satiated, recreating previous observations of the relationship between gregarisation and foraging [46,47].

Finally, we consider the effect of hunger on group dispersal. In these simulations we take a mass of locusts, $M$ from 1 to 10, with an initial gregariousness of $n_g = 0.95$ both with and without a hunger state dimension (when hunger is included $n_h = 0.95$ initially). We place them at the centre of a domain without food, and run the simulation to $t = 10$. We then measure the peak density of the locusts and the mean gregariousness at $t = 10$. The results can be

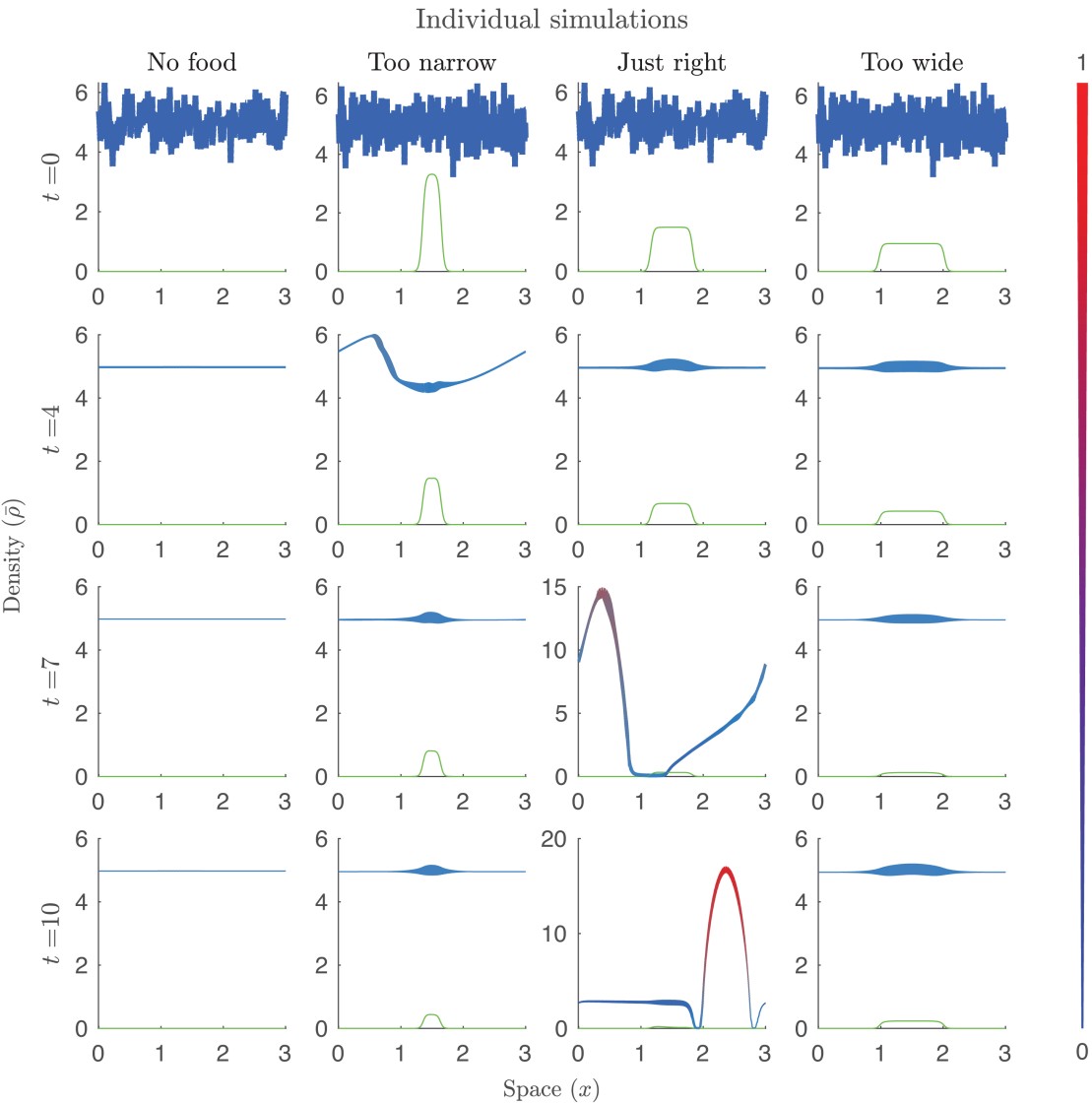

**Fig 4. Individual simulations with varying food footprints.** In the plots, food is plotted in green with the total locust density given by a line that varies in colour based on the average locust density at that point, changing from blue being fully solitarious to red being fully gregarious. For the simulations, ambient locust density is $\rho_{amb} = 4.95$, $\omega = 0\%, 10\%, 22.5\%, 35\%$, and $t = 0, 4, 7$, and 10.

seen in Fig 5. We find that hunger acts to decrease both the final peak density and the mean gregariousness, leading to a faster breakdown in group cohesion. In the absence of hunger, the locust group dispersed when $M \leq 5$, whereas with hunger we saw dispersal when $M \leq 6$. These results, when considered in conjunction with the group heterogeneity results, suggest a simple mechanism for small group dispersal. Hungry locusts move towards the group edges, become increasingly solitarious, and then leave the group. As the whole group becomes hungrier, density decreases, leading to a decrease in gregariousness. This in turn further reduces

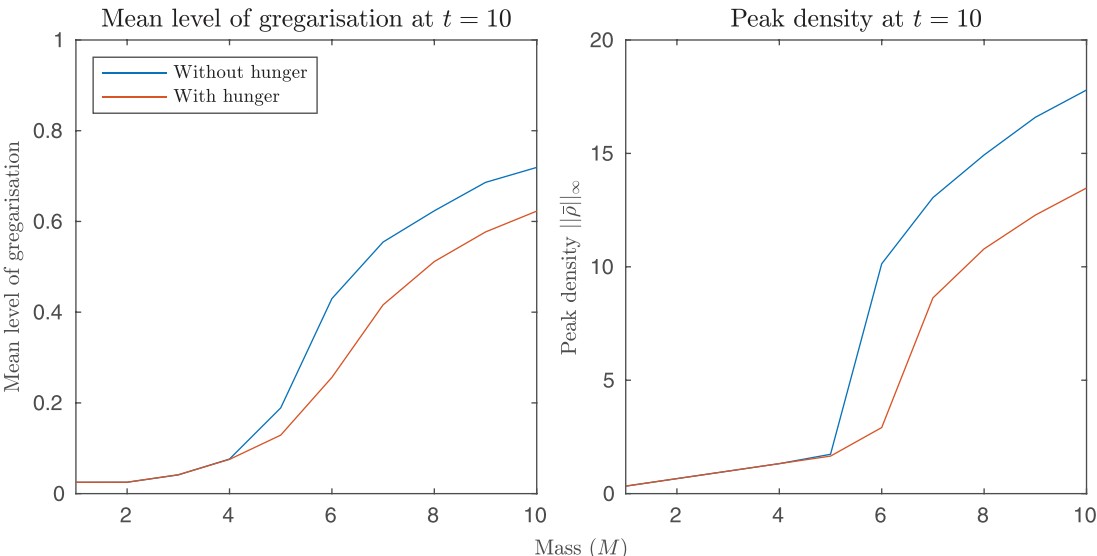

**Fig 5. The effect of hunger on locust dispersal.** In the left plot, is the average gregariousness with respect to mass and the right plot is the peak density (at time $t = 10$). In both plots, the red line includes hunger while the blue line does not. We can see that hunger has lead to a greater loss of gregariousness and a reduction in the peak density thus leading to an increase in locust dispersal.

density, which further reduces gregariousness, etc. When we combine this with the previous result of decreasing maximum density with deceasing food footprint, we hypothesise that groups formed in sparser food environments may be initially less stable than those formed in environments with more even food distributions.

## 4. Discussion

In this paper, we have presented a new continuum model for self-organised collective movement, which takes into account the internal states of individuals, such as their level of hunger, as well as their interactions with their environment. Existing mathematical models typically assume a homogeneous population, in the interests of tractability, but we know from observations and experiments that population heterogeneity plays an important role in group structure and dynamics [48]. We then applied our model to investigate locust foraging and group structure with state dimensions of hunger and gregarisation.

Despite the increased complexity introduced by including heterogeneity, it was still possible to gain some useful insights through analysis and numerical simulation of the model. Using the method of gradient flows, we found that hunger acts to decrease the maximum density of locusts by increasing their dispersal while the presence of food has the opposite effect (this relationship is inverted for the size of the aggregation's footprint or support). Secondly, increasing gregarisation and attraction strength increases the maximum density of locust groups. Through a linear stability analysis of a single state we found that food decreases the required gregariousness (and by extension density) for group formation and hunger increases it. These conditions carry over if the locusts can be considered as occupying two discrete states, with the fraction in the more gregarious state as the gregarious mass fraction. We saw that as the solitarious locusts become hungrier (or more solitarious) this increases the gregarious mass fraction required for group formation, with the presence of food acting to decrease it. Next, if the gregarious state has less local repulsion than the solitarious state this decreases

the gregarious mass fraction required for group formation and vice-versa, i.e., if all the gregarious locusts are satiated while the solitarious are not, this would again decrease the mass fraction required. In addition, as the steady-state density ($\rho_c$) increases (or approaches zero) the gregarious fraction required for group formation increases suggesting an upper (and lower) locust density in order to form groups. In contrast to previous studies, the aggregation process can be brought about not only by increasing the gregarious mass fraction but by changing the internal state or environment of the locusts. For example, given a population that is below the critical mass fraction for group formation, if it becomes more satiated (leading to a decrease in repulsion behaviours) this could trigger the locusts to form aggregations. Conversely, decreasing satiation or gregarisation could prevent aggregations forming. Finally, many of our results are derived in a more general sense in S2 Appendix and are thus applicable to organisms other than locusts.

Our numerical results from Sect 3.2 showed that considering internal states has a variety of effects on group formation and heterogeneity. When looking at the effect of food on group formation, including hunger vastly decreases the required density for group formation and gives rise to an optimal food width. Specifically, when the food footprint is too narrow there is attempted group formation, but there is an insufficient mass of gregarious locusts (or they are insufficiently satiated) so the group does not persist. Alternatively, if the food footprint is too wide no attempt at group formation occurs. Furthermore, decreasing the food footprint resulted in a decrease in the final maximum density of the resulting group.

Within the group structure itself, we saw that once a group has formed, on average the most gregarious and satiated locusts end up in the middle of the group with a reduction in gregarisation and an increase in hunger towards the edges. Our results with modelling hunger suggest a rather simple mechanism for this observed behaviour. Hunger increases the strength of local repulsion, which drives the hungry individual to the edges of the group and has the additional effect of lowering the local maximum density. This, in turn leads to a decrease in gregariousness. When this occurs with a sufficiently small mass of locusts it leads directly to group dispersal. This effect, combined with the decrease in peak density with decreasing food footprint suggests that groups formed in sparse food environments may be less stable than those formed in more evenly distributed food environments. However, we note that we have not included hunger dependent cannibalism which may operate to counteract this effect. Finally, we found that the most gregarious locusts appear to be the most satiated, highlighting that gregarisation offers an advantage when foraging in patchy food environments [8,46,47,49].

Our model includes some simplifying assumptions that limit its direct biological relevance. In addition, in order to perform the numerical experiments, we needed to limit the number of state variables due to computational complexity with one explorable avenue being more efficient numerical methods. Further explorations within the context of locust foraging could look at changing our assumption that the effect of internal states and the environment were separable and relatively simple, or could add more states and interactions. Some examples for extension are: more complex hunger and feeding interactions [19] including cannibalism [50], or the inclusion of a heterogeneous age structure. In addition, the no-flux boundary conditions on the state variables could be changed to allow locusts to 'leave' the state-space, this could be used to model mechanisms such as death by starvation. Finally, while we have focused on locusts and foraging, the modelling technique itself and many of our results have been derived generally and can be used to explore a variety of different scales from microscopic to macroscopic [29,33].

## Supporting information

**S1 Appendix. Model derivation.** The full detailed derivation of the flux terms given in the model section.
(PDF)

**S2 Appendix. Full analytic results.** The full detailed derivations of the analytic results given in the PDE model analysis section for the more general model.
(PDF)

**S3 Appendix. Numerical scheme.** The full detailed derivation of the numerical scheme used for simulating the numerical results as well as some extra numerical results.
(PDF)

**S4 Appendix. Parameter estimation.** Derivation of parameters given in the simulation section.
(PDF)

## Author contributions

**Conceptualization:** Fillipe Georgiou, Camille Buhl, J.E.F. Green, Bishnu Lamichhane.

**Formal analysis:** Fillipe Georgiou.

**Funding acquisition:** Ngamta Thamwattana.

**Investigation:** Fillipe Georgiou.

**Methodology:** Fillipe Georgiou, J. E. F. Green.

**Software:** Fillipe Georgiou, Bishnu Lamichhane.

**Supervision:** Camille Buhl, J. E. F. Green, Bishnu Lamichhane, Ngamta Thamwattana.

**Visualization:** Fillipe Georgiou.

**Writing – original draft:** Fillipe Georgiou.

**Writing – review & editing:** Fillipe Georgiou, Camille Buhl, J. E. F. Green, Bishnu Lamichhane, Ngamta Thamwattana.

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
