## [Decision Letter · Decision Letter 0]

3 Jan 2024

Dear Dr Georgiou,

Thank you very much for submitting your manuscript "Including organism and environmental heterogeneity in kinematic continuum models of collective behaviour with applications to locust foraging and group structure" for consideration at PLOS Computational Biology.

As with all papers reviewed by the journal, your manuscript was reviewed by members of the editorial board and by several independent reviewers. In light of the reviews (below this email), we would like to invite the resubmission of a significantly-revised version that takes into account the reviewers' comments.

I found your work very interesting and agree with all reviewers that introducing individual-level heterogeneity in continuous models is an important step forward in the collective behavior literature and in theoretical ecology broadly. However, I also agree with the Reviewers that, in its current form, the submission lacks focus on which are the relevant biological insights it provides. Also, given the broad readership that could be interested in understanding your framework to introduce individual-level heterogeneity in continuous models for different systems, I would encourage you to work hard on making the text more accessible for non-experts in locust biology and locust modeling. When preparing your revised submission, please respond appropriately to all Reviewer comments, but I encourage you to pay special attention to the two points above.

We cannot make any decision about publication until we have seen the revised manuscript and your response to the reviewers' comments. Your revised manuscript is also likely to be sent to reviewers for further evaluation.

Sincerely,

Ricardo Martinez-Garcia

Academic Editor

PLOS Computational Biology

James O'Dwyer

Section Editor

PLOS Computational Biology

Reviewer's Responses to Questions

**Comments to the Authors:**

Reviewer #1: Comments are in the attached pdf file.

Reviewer #2: Comments to the Authors

The manuscript addresses the formation of dense regions by moving organisms which are individually characterised by a state vector that can evolve in time depending on local conditions, somewhat similarly to some chemical reactions between molecules. The authors devise a kinetic field theory of the reaction-advection-diffusion type. The effective force contributions are controlled by the current internal state and by a spatial distribution of environmental conditions such as food availability, temperature, etc. Some model features are discussed and numerical simulations performed, with applications to locust swarms. "Phase" separation into dense and dilute regions occurs if the overall density is within a certain range while a non-local attraction is sufficiently strong, a local diffusion is sufficiently low and a local repulsion is sufficiently low as well. For locusts, the authors find that more attractive (called gregarious) and less hungry individuals (where the level of gregariousness and of hunger define the internal state) will typically stay nearer the centre of the aggregates. For some conditions, agglomerates do not form even if initially attempted in the dynamics.

The motivation is certainly valid since combining two internal state dynamics is expected a priori to generate important modifications with respect to previous results by the same group in the same journal. Particularly interesting is the case where the initial conditions induce agglomeration but food is consumed faster than gregarization or hunger is maintained (as I interpreted it), therefore generating failed attempts to agglomerate. At the same time, the problem is clearly relevant to biology.

However, in my opinion, the manuscript has a large number of problems and the relevance of the final set of results is unclear despite the relevance of the model itself. I personally found that the text is poorly written. Many variables are not appropriately introduced, while modelling and (some important) parameter choices are not discussed, at least not in the main text. Having to go to the appendices multiple times is a problem. A large number of sentences are way more intricate than needed. Commas have been used as final periods (and vice versa) dozens of times, confusing the flow. Importantly, the rationale behind many modelling choices and results are not attempted to be explained heuristically, but rather just given, leaving the ultimate interpretation job to the reader. Also, I feel like the paper would have been much clearer if it started with the locust example straight away. Finally, the take-home novelty would be easier to grasp if the authors stated more clearly from the outset that the new ingredient with respect to their previous model is specifically the hunger dynamics.

Regarding the relevance of the results, although there are certainly a few insights which are significantly easier to obtain through the simulations, many others can be too easily anticipated only from stating the chosen model. There are not many quantitative results that can be used to obtain new unanticipated insights (the linear stability analysis is poorly performed; more details below). In fact, many insights highlighted by the authors are the same as in many other phase separation problems. In summary, although I do believe there are important insights, they get muddled in the middle of many repeated trivial results. On top of that, it is not completely clear which of the insights/results are actually new. By introducing hunger dynamics, individuals that get satiated move less and thus have a higher tendency to agglomerate and this can further induce agglomeration. Is that all? I think there is a bit more, but the punch line or take-home message is not clear at all, particularly in the abstract.

Therefore, I would recommend the text to undergo at least major revision. Below I raise many particular points, issues and suggestions, to try and help the authors with the process.

- Regarding the title/abstract, I'd understand it faster if it said "population" heterogeneity. The authors may also want to consider adding some word like "dynamic" or "time-evolving", so that it says "population and environmental dynamic heterogeneity".

- When the authors say "complex behaviors arise from simple … interactions", they may want to say "relatively simple".

- Just a curiosity, models of movement coupled to internal dynamics have also been called "swarmalators" in the nonlinear dynamics physics community.

- For the abstract, authors may want to explain that gregarious means more sociable.

- Sentences like "finding that density and formation is most affected by the ratio of attractive to dispersive interactions." sound too obvious. Perhaps the authors can drop the word "finding" since this kind of behavior is old news and widespread in many models in the literature.

- First order SPPs can also capture alignment and pursuit. This division seems artificial. The authors can just say that inertia can be ignored for many animal movement problems.

- All equations should be numbered. For the first equation (not numbered), the authors could just explain the mathematics of their modelling of the non-local contribution to the velocity as a convolution of a function (which, I don't know why, is expressed in terms of the grad of another function Q). Something like "We assume that each individual decides on its velocity based on a weighted average of the organism density in a neighbourhood around it, with closer individuals having higher weight." should suffice.

- There are sentences which are identical to those of Ref. [22].

- Regarding the sentence "this relationship might be more continuous", only later I understood that "relationship" here meant the level of gregariousness.

- When "environment" is first mentioned, one should already give the food availability example. It remained ambiguous before that.

- For the term with rho◯2, the authors could explain that this corresponds to a flux velocity proportional to the negative of the gradient of rho◯2 since individuals want to move away from where collisions are most likely, like a repulsive autochemotaxis. The kind of explanation is necessary (I needed to look at other references). Besides, with that, no new microscopic derivation would be needed.

- Comparing the first unnumbered equation and Eq. (4), it seems that the gradient is taken after convolution only in the second case. Which is which and why?

- Combining Eq. (6) and the unnumbered equation just below it, there seems to be a repeated term.

- For section 3, the authors should say in its title that it applies only to the simpler case without their dynamic heterogeneity.

- In "we will approximate the support of an aggregation", the authors should explain the new capital omega (different than the omga prime) more carefully.

- The sentence "Finally, any change in state that increases…" is covered by the "vice versa" just before, right? If so, eliminate the repetition.

- Explain that function B in (14) is a special function so we don't need to go to the appendix to check whether it is a new thing not defined before.

- When the authors say "the dotted lines represent the minimum estimates", explain why the minimum is taken.

- For the 98% analysis, other percentages perhaps give somewhat different results. Is this value the one that gives the best comparison? Perhaps the authors could comment on that. In any case, I understand the idea here and that no big difference should arise for other percentages. No big deal.

- When n_2 is introduced in section 3.1.3, it seems the authors are calling n_2 the lever of hunger and thus more hunger means less diffusion. Is it not the level of satiation? This becomes even more confusing when n_2 becomes denoted by n_h below.

- Regarding the notation of some functions f_n or f_l, in some cases the E is kept in the list of dependent variables even when it is not there while in some other similar case it is not.

- The modelling choices for f_n, f_l, f_c, etc. have to be justified heuristically. For instance, for the easiest one, a sentence like this would suffice: "Here we assume that the hungrier individuals decide to move more, which can be understood as a mechanism to find food quicker." or similar. Other cases are less obvious and should be discussed. These choices directly control final results. In fact, many final results can already be anticipated from these choices. They need to be justified. Just saying that a previous paper used it is not enough. Give at least a taste of why it is chosen like that.

- Title and caption title of Fig. 2 do not seem accurate.

- rho_c is introduced without definition in the main text. The same for the Fourier transform of Q.

- For the two states analysis, s and g mean solitarious and gregarious. This should be said explicitly.

- Regard using B below (20), notice that B was already used for the special function.

- In (20), phi_g bar is different than phi_g star. Explain the difference.

- F_c, lambda(n) and psi(n) are introduced without definition in the main text.

- Fig. 3 is not referred to in the main text. It just appears. By the way, is the y axis n_g? Please say it. Also give the number for the curve called "maximum" so that it can be compared to the others.

- The quantitative analytic results from the linear stability analysis (for instance, the transition values) are not compared to simulations, right?

- Regarding (19), notice that Q also appears in the denominator. So taking only k=0 is not enough to identify the maximum lambda, right?

- What is the difference between rho_amb and rho_c/2?

- After (37), explain that this shape is like a patch/plateau of food concentration surrounded by no food

- The sentence "food decreases the required density… as opposed to optimal width" was very confusing.

- In "Figure 5 we can see both" there is no "both" there.

- I do not think using thickness of the line as a plotting technique was a good choice. It is not sufficiently clear.

- Explain heuristically the re-emergence of optimal width effect. I believe this kind of insight is what will make the paper valuable.

- There are some typos. Please use a checker.

- In the discussions, rho bar is used as a parameter when it is actually a variable (the authors probably meant rho_c or rho_inf).

- I would not use the word "support". This is unnecessary mathematical jargon for the bio community. Just say the size of the patch or agglomerate or similar.

- There are long parts in the appendices which are almost identical repetitions from other parts.

- The result described in "When the food is too narrow there is attempted" is the most insightful. I would suggest putting a focus on this.

Reviewer #3: This study takes a class of previously explored locust models (by the authors and many others) and adds an internal state variable to model heterogeneity with an eye on locust biology (notably modeling hunger and degree of gregariousness). The models here are well motivated but by adding a state space to these models one observes an explosion of parameters and unknown functions and essentially increases the dimensionality of independent variables in the model significantly increasing the complexity.

The models here are challenging (dare I say approaching unwieldy) but the authors skillfully finesse some insight from them.

This reviewer believes that the authors should be praised for exploring this territory and believes that studies incorporating heterogeneity akin to what is done here are a path forward in continuum modeling of swarms.

The writing of the paper assumes (in this reviewer's opinion) a significant familiarity with past work in this field which limits its accessibility. It could use a thorough editing with an eye on readability for a broader audience. The comments below point out a number of things that struck this reviewer that could be elucidated.

Line 42: “. . . a the locust . . . “

Lines 72-75: Could be written clearer and with more specificity:

• Line 74: “. . . . local interactions are repulsive. . . “ What about gregarization? Isn’t this a local interaction?

• Line 75: Why does 4 call out “any state” but 3 does not? Also is the state of the target organism, or the influencing organism (or both) that matters/doesn’t matter?

Lines 78-84: At least some comment should be made here about continuous/discrete states. As an example many previous models have made solitary/gregarious a binary state. Is this allowed in this model or not? Also, a discrete model is introduced later on - could this be foreshadowed?

Line 80-81: Why not define $\Omega_n$ here as opposed to line 85? Also is there any difference between “state space” and “complete state domain”.

Line 87: The definition here of $E$ is vague – is a vector of variables (food density, sunlight, etc)? Do they depend on space? time?

Line 93: The use of “a movement” here is confusing (nothing is moving) – change or evolution might be clearer.

Line 94: There is a subtlety in the flux equation I’d encourage the authors to call out explicitly – this form means that the state of a locust evolves continuously (i.e. it doesn't jump between states) in time (assuming the fluxes are not infinite). I believe this is the intent of the author and it should be made clear here or earlier in the text.

Line 101: While D is a diffusion constant, gamma here most certainly is not. It is a advective flux mediated by the function $\tau(\bar{\rho})$ introduced in the appendix. I’d also say that f_l(n,E) is a mobility. In any case, the authors should spend a few lines explaining biologically what these terms here. I think the first term is diffusion of organisms where the diffusion constant is mediated by the environment and the organism state and the second term is an advective flux along gradients away from concentration densities – but the authors can surely do a better job of giving intuition about what is going on here. Along these lines talking about units for the constants may be useful here.

Line 106: Might it be better to say the social potential is a function of distance between organisms and independent of location, state and time. The use of the word “space” here is ambiguous.

Line 106 and 109: It seems to me that only the strength of the social potential is mediated by the internal state and the environment. I’m not even sure what is meant by “non-local force” here.

Line 110: I’d encourage the authors not to use the term “velocity” here or to explain it is an analogy here. Even before lunch, I would be unlikely to say my hunger is increasing at 30 meters/second.

Line 133: The manuscript says “all the organisms are in the same unchanging state” the description of which is rather vague (I think unchanging here means relative to time but even that is unclear to me) – I think the authors want an N-dimensional delta function here for the state vector, but part of the problem goes back to line 113 because we have no idea if the state could evolve in time. I believe the authors want to say something like “suppose the state vector is constant and that $v_n=0$ so that the state remains constant for all time.” Ideally there would be a topic sentence earlier in section 3.1 that tells the reader that this approximation is being made. This also will need to be made consistent in Appendix B

Line 135-136: The evolution equation here has a long history and the author should give a few references. Notably as the equation is introduced here they should note that $\bar{\rho}=0$ OR the variation in energy (after 137) is zero. Considering the large mass solutions that are discussed soon after have compact support this discussion should foreshadow this. Also, strictly speaking the equation after 137 only defines a steady state (not necessarily a minimizer).

Line 206: Presumably the subscripts g and s are nods to gregarious and solitarious but this is not spelled out for the reader – a topic sentence about this at the beginning of 3.2.2 would be useful.

Line 239: The term “separable” here should be defined mathematically and described biologically.

Section 4: The authors should explain why the particular range of locust densities is used and why figure 4 and 6 have that density decreasing with vertical y (the authors may make this choice if they have a good reason but they should warn the reader).

**Have the authors made all data and (if applicable) computational code underlying the findings in their manuscript fully available?**

Reviewer #1: Yes

Reviewer #2: None

Reviewer #3: Yes

PLOS authors have the option to publish the peer review history of their article (what does this mean?). If published, this will include your full peer review and any attached files.

Reviewer #1: No

Reviewer #2: **Yes: **Pablo de Castro

Reviewer #3: No
---

## [Decision Letter · Decision Letter 1]

13 Jul 2024

Dear Dr Georgiou,

Thank you very much for submitting your manuscript "Including dynamic population and environmental heterogeneity in continuum models of collective behaviour with applications to locust foraging and group structure" for consideration at PLOS Computational Biology. As with all papers reviewed by the journal, your manuscript was reviewed by members of the editorial board and by several independent reviewers. The reviewers appreciated the attention to an important topic. Based on the reviews, we are likely to accept this manuscript for publication, providing that you modify the manuscript according to the review recommendations.

Sincerely,

Ricardo Martinez-Garcia

Academic Editor

PLOS Computational Biology

James O'Dwyer

Section Editor

PLOS Computational Biology

Reviewer's Responses to Questions

**Comments to the Authors:**

Reviewer #1: I think the new structure now provides a better reading flow and a clear story.

More importantly, the message is there and easily accessed.

Therefore, I am recommending the current version for publication, but as a final touch, I suggest the authors to improve minor details in the figures if this still possible. Fonts are a bit everywhere, sometimes it could be too small to read in the 100% view. Line thickness and better color choice could also help in visualization of the results.

Reviewer #2: Comments to the Authors - After 1st round of reviews

First, I would like to thank the authors for their revised version. I have carefully reviewed the entire manuscript again. In my opinion, this new version is a significant improvement but still has some issues, primarily related to presentation (see below). In particular, I think the authors should have focused more on the optimal food patch size effect, which is non-trivial and not straightforward to predict only by knowing the model ingredients.

Nonetheless, I am happy to recommend the paper for publication after minor revision, without the need for me to review the manuscript again.

Minor issues

- I think the new title is not clear yet. I suggest the previous version I had recommended or "Including populational and environmental dynamic heterogeneities in continuum models of collective behaviour: Applications to locust foraging and group structure". The authors include two types of heterogeneities which are dynamic. So it makes sense to use the plural. I would not put dynamic before population and environmental as this gave the title another meaning.

- Many model assumptions are given without justification. One has to stop and think what are the potential implications and convince oneself that they are not too problematic.

- Typo: often affects -> often affect

- The abstract could indicate somehow that gregarisation is time-dependent.

- Why cannot environmental interactions be non-local too? I sense food far from me and I move towards it, similarly as I sense high locust density far from and move accordingly.

- I do not understand what "and not by those of the target" meant there.

- The first time in the abstract you mention gregarisation, it is not explained. Then you do explain. Maybe invert that.

- In the abstract, in "We also note … hunger.", do you mean optimal food patch size? You said only optimal food. Also, you may want to say this phenomenon emerges due to hunger effects. The writing was a bit unclear to me at first.

- the ratio of non-local to local -> the ratio of local to non-local

- Sentences like "is likely to form" are strange as the model is deterministic. The small stochasticity on the initial conditions should not play a role in that sense.

- By empty space you mean space without food, right?

- You replied that rho_amb and rho_c/2 are the same. Is this said in the text?

- The sentence part "and with hunger being the key component…" seems to be missing something.

- In your reply, when you say you changed to steady-state density to avoid the use of symbols, I would actually use symbols to help clarify, on top of using words.

- "mechanism for the dispersal of locust groups" gives the impression that the whole group is cohesively dispersing across space. Maybe say group disintegration.

- In the intro, when you say second order models are capable of capturing alignment, this is not specific to second order models. In fact, most models in active matter do not consider interia and yet consider alignment, etc. Then you say first order models are used for another class of problems. I don't think you are dividing the use of first and second order models adequately.

- Typo: using SPP models -> using SPP models exist

- under the assumption of homogeneity -> under the assumption of environmental heterogeneity

- When you introduce E, it becomes clear only much later that E depends on space. This confused me at some point.

- Although I had suggested mentioning collisions, perhaps the best word is close encounters. But that's up to you. I am not sure.

- Why do you write the kernel as grad Q and not just a kernel K? And why is f inside grad in (7)? Does it make a difference?

- Above (14), it felt like you forgot to choose f_hl, but then it appears later.

- n_h is the level of satiation and not the level of hunger, but the subindex h indicates otherwise. Perhaps say it explicitly to avoid confusion.

- Below (19), I'd say "completely starving (n_h=0)" and then "completely satiated (n_h=1)".

- Explain difference between lambda and psi

- When you say "We begin by using gradient flow methods", perhaps say that this is to numerically solve your PDE, and also say that here you are also using analytical methods. Or, if you are using the expression gradient flow methods for both numerical and analytical, make that clear.

- "label the footprint of this aggregation as the support" is not a clear enough sentence. Why not "label the size of this aggregate as"?

- Ensure you define/write rho_inf as M/Omega

- You do not say how you find (22). Mention Appendix or hint that you are plugging your assumptions for the rectangular aggregate into the PDE.

- comparing them to simulation results -> comparing them to simulation results obtained by numerically integrating Eq. (x)

- At some point you mention a pseudo steady state. I wonder whether your true steady state could be formed by many aggregates as in other problems of non-local or non-linear diffusion.

- Above the paragraph of (28), there is "Appendix ??"

- Why do you introduce eps squared terms in the linear stability analysis if you do not keep them?

- At the beginning of subsection 3.2 Simulations, mention the appendix on parameter estimation.

- You use real parameters with units to estimate your model parameters, but then they have no units.

- Typo: Tables 1 and2 -> Tables 1 and 2

- "for a total of 10 hours." This is model time, not computational time, right?

- There is a ?? on table 2

- From "In order to ensure that simulations …" until the end of the paragraph, I did not understand the point you want to make or the logic behind your analysis.

- being the initial food footprint -> being the initial food footprint size?

- Just after (34), perhaps mention that there will be later plots of (34)

- When you say "likely due to no symmetry…", you are saying that, without food, there is no group formation. But is this not only a matter of parameters? There is group formation in other problems of non-linear/non-local diffusion where "symmetry breaking" is spontaneous and the environment is homogeneous. Anyhow, if you can have group formation without food, then your conclusions about the optimal food patch size apply only when there is no group formation in the small/absent food patch case.

- You should add an arrow in Fig. 3 to highlight even more that density increase goes down.

- The paragraph above Fig. 3 is confusing. I understood it but it took me more than needed.

- Font sizes on figures are too small.

- About Fig. 5, is this for M and food distributed in space in which way?

- You say n_h=0.95 initially with hunger dimension, but before you said that n_h=0.5 without hunger dimension. Should not n_h=0.5 too with hunger dimension (and let it evolve) for a fair comparison? This may be particularly relevant because t=10 only.

- affect of food -> effect of food

- When you say that gregarisation offers an advantage because they are more satiated, is it not the other way around, i.e., they are more satiated and thus move less, form aggregates and gregarise?

Reviewer #3: I will attach my review as a .pdf

**Have the authors made all data and (if applicable) computational code underlying the findings in their manuscript fully available?**

Reviewer #1: Yes

Reviewer #2: Yes

Reviewer #3: Yes

PLOS authors have the option to publish the peer review history of their article (what does this mean?). If published, this will include your full peer review and any attached files.

Reviewer #1: No

Reviewer #2: No

Reviewer #3: No

Figure Files:

Data Requirements:

Reproducibility:

References:

---

## [Editor Report · Decision Letter 2]

9 Feb 2025

Dear Dr Georgiou,

We are pleased to inform you that your manuscript 'Including population and environmental dynamic heterogeneities in continuum models of collective behaviour with applications to locust foraging and group structure' has been provisionally accepted for publication in PLOS Computational Biology.

Best regards,

Ricardo Martinez-Garcia

Academic Editor

PLOS Computational Biology

James O'Dwyer

Section Editor

PLOS Computational Biology

---

## [Editor Report · Acceptance letter]

PCOMPBIOL-D-23-01370R2

Including population and environmental dynamic heterogeneities in continuum models of collective behaviour with applications to locust foraging and group structure

Dear Dr Georgiou,

I am pleased to inform you that your manuscript has been formally accepted for publication in PLOS Computational Biology. Your manuscript is now with our production department and you will be notified of the publication date in due course.

With kind regards,

Anita Estes
